# Amyloidogenic Peptides: New Class of Antimicrobial Peptides with the Novel Mechanism of Activity

**DOI:** 10.3390/ijms23105463

**Published:** 2022-05-13

**Authors:** Oxana V. Galzitskaya, Stanislav R. Kurpe, Alexander V. Panfilov, Anna V. Glyakina, Sergei Y. Grishin, Alexey P. Kochetov, Evgeniya I. Deryusheva, Andrey V. Machulin, Sergey V. Kravchenko, Pavel A. Domnin, Alexey K. Surin, Viacheslav N. Azev, Svetlana A. Ermolaeva

**Affiliations:** 1Institute of Protein Research, Russian Academy of Sciences, 142290 Pushchino, Russia; st.kurpe@gmail.com (S.R.K.); panfilov.alexander@mail.ru (A.V.P.); quark777a@gmail.com (A.V.G.); syugrishin@gmail.com (S.Y.G.); alan@vega.protres.ru (A.K.S.); 2Institute of Theoretical and Experimental Biophysics, Russian Academy of Sciences, 142290 Pushchino, Russia; 3Institute of Mathematical Problems of Biology, Russian Academy of Sciences, 142290 Pushchino, Russia; 4Keldysh Institute of Applied Mathematics, Russian Academy of Sciences, 125047 Moscow, Russia; 5The Branch of the Institute of Bioorganic Chemistry, Russian Academy of Sciences, 142290 Pushchino, Russia; rambrent@mail.ru (A.P.K.); viatcheslav.azev@bibch.ru (V.N.A.); 6Institute for Biological Instrumentation, Federal Research Center “Pushchino Scientific Center for Biological Research of the Russian Academy of Sciences”, 142290 Pushchino, Russia; janed1986@ya.ru; 7Skryabin Institute of Biochemistry and Physiology of Microorganisms, Federal Research Center “Pushchino Scientific Center for Biological Research of the Russian Academy of Science”, 142290 Pushchino, Russia; and.machul@gmail.com; 8Institute of Environmental and Agricultural Biology (X-BIO), Tyumen State University, 625003 Tyumen, Russia; svkraft@yandex.ru; 9Gamaleya Research Centre of Epidemiology and Microbiology, 123098 Moscow, Russia; paveldomnin6@gmail.com (P.A.D.); drermolaeva@mail.ru (S.A.E.); 10Biology Faculty, Lomonosov Moscow State University, 119991 Moscow, Russia; 11State Research Center for Applied Microbiology and Biotechnology, 142279 Obolensk, Russia

**Keywords:** antibacterial peptides, amyloids, co-aggregation, membrane, penetration, amyloidogenic peptides

## Abstract

Antibiotic-resistant bacteria are recognized as one of the leading causes of death in the world. We proposed and successfully tested peptides with a new mechanism of antimicrobial action “protein silencing” based on directed co-aggregation. The amyloidogenic antimicrobial peptide (AAMP) interacts with the target protein of model or pathogenic bacteria and forms aggregates, thereby knocking out the protein from its working condition. In this review, we consider antimicrobial effects of the designed peptides on two model organisms, *E. coli* and *T. thermophilus*, and two pathogenic organisms, *P. aeruginosa* and *S. aureus*. We compare the amino acid composition of proteomes and especially S1 ribosomal proteins. Since this protein is inherent only in bacterial cells, it is a good target for studying the process of co-aggregation. This review presents a bioinformatics analysis of these proteins. We sum up all the peptides predicted as amyloidogenic by several programs and synthesized by us. For the four organisms we studied, we show how amyloidogenicity correlates with antibacterial properties. Let us especially dwell on peptides that have demonstrated themselves as AMPs for two pathogenic organisms that cause dangerous hospital infections, and in which the minimal inhibitory concentration (MIC) turned out to be comparable to the MIC of gentamicin sulfate. All this makes our study encouraging for the further development of AAMP. The hybrid peptides may thus provide a starting point for the antibacterial application of amyloidogenic peptides.

## 1. Introduction

There is a lot of evidence for the biological activity of proteins and peptides that exhibit the ability to aggregate [1,2,3,4,5,6,7,8]. There is some relationship between the amyloidogenic and antimicrobial properties of peptide(s), the reasons for which have not been fully known [9]. The majority of antimicrobial peptides have a membrane-active effect. It is known that the positive charge, hydrophobicity, and amphipathicity of AMPs play an essential role in the interaction with the membrane [10]. It is assumed that AMPs can form pores in membranes by three main mechanisms of pore formation. The Barrel stave model assumes the oligomerization of perpendicularly oriented AMP molecules inside the membrane [11]. In contrast to the Barrel stave model, toroidal pore formation occurs upon binding to lipid heads [12,13]. The carpet model assumes a parallel orientation of the AMP to the membrane plane, which leads to the formation of micelles and the destruction of the membrane [14]. Similarly, bacterial lysis can occur when AMPs interact with the membrane surface; classical cationic antimicrobial peptides, such as LL-37 [15] and NK-2 [16], and amyloidogenic antimicrobial peptides, such as Aβ peptide [17,18] and human islet amyloid polypeptide hIAPP [19].

In general, interaction with the membrane is characteristic of amyloidogenic proteins and peptides, regardless of their structure and function of the native molecule, due to a large number of hydrophobic amino acid residues [20]. A study of the enantiomeric forms of the amyloidogenic peptides baboon θ-defensins (BTD-2), protegrin-1 (PG-1), and polyphemusin-1 (PM-1) showed that the activity of the chiral forms of the peptides does not change, which indicates an act via binding to bacterial membranes [21]. The antimicrobial amyloidogenic peptide uperin 3.5 from an Australian amphibian showed the ability to bind to both eukaryotic and bacterial membrane models [4]. The authors emphasize the importance of a positive charge and a moderate tendency to aggregation for the membrane activity of the peptide. The model of interaction between uperin 3.5 and the membrane depends on the composition of the membrane. The action on the eukaryotic membrane model stimulates the formation of pores, while the interaction with bacterial models leads to disruption (carpet or detergent effect) of the membrane. Numerous studies on the membrane activity of Aβ peptide, Tau, and hIAPP show that adhesion on the membrane promotes the formation of an a-helical conformation and oligomerization of molecules, which can lead to the appearance of ion-permeable pores or disrupt the lipid bilayer and can lead to disruption of the ionic balance and cell death [20]. As well as uperin 3.5, Aβ peptide can lead to membrane disruption when the membrane composition changes. Sciacca et al. showed a two-stage mechanism of membrane disruption. In the second stage, fibrilization of Aβ peptide leads to a detergent-like mechanism to fragment the lipid bilayer [18,22]. Agglutination of bacteria can be considered as another antimicrobial mechanism of amyloidogenic peptides. Some forms of Aβ peptide and Eosinophil Cationic Protein result in the death of microorganisms and the clumps of bacteria after treatment [7,23]. Using the binding model of the antimicrobial peptide thanatin-LPS, it was shown that the interaction of the peptide with the cell surface is accompanied by binding to polar regions modulated by ionic and hydrogen bonds [24]. This is followed by immersion and the formation of the β-hairpin structure. Interaction between cell surfaces occurs when constructing an antiparallel N-terminal β-strand fold of two thanatin subunits. It is noted that agglutination can not only accompany but be a key factor for the manifestation of antimicrobial activity, and amyloidogenicity is an essential factor determining cellular toxicity.

Some peptides may have a non-lytic mechanism of action. The antimicrobial activity of peptides may be associated with inhibiting the biosynthesis of DNA, RNA, proteins, and other vital processes, such as buforin II and indolicidin dermaseptin [25,26,27]. Amyloidogenic peptides also show activity against intracellular targets. It has been observed that oligomeric forms of amyloidogenic proteins and peptides often contribute to cytotoxic effects [20]. Impaired proteostasis in Alzheimer’s Disease, Parkinson’s Disease, Type II Diabetes, and Amyotrophic Lateral Sclerosis is usually associated with abnormal protein folding, which leads to aberrant interactions of both amyloidogenic proteins and peptides with each other, and the targets with which amyloids interact can contact [28,29]. It has been shown that Aβ peptide inhibits Akt1 phosphorylation, thereby inhibiting the Insulin/Akt Signaling Pathway in human neurons [30], which may negatively contribute to neuronal viability. Thus, amyloidogenic peptides and proteins exhibit a wide range of cytotoxic and antimicrobial effects based on both membrane-active and intracellular mechanisms of action.

Several groups of authors, including us, have developed variants of amyloidogenic peptides that exhibit antimicrobial activity and whose main purpose is intracellular targets. Bednarska et al. proposed the design of AMPs based on the ability of aggregation-prone regions (APRs) to homologous aggregation [1]. According to their idea, targeted aggregator peptides induce specific aggregation of bacterial proteins and display strong bactericidal effects. Using the TANGO algorithm [31], APRs in the bacterial proteome were selected. Their design suggests duplicate sequences of APRs separated by a glycine–serine linker and flanked by arginine and lysine to maintain colloidal stability (Figure 1). A similar approach was used by Khodaparast et al. to develop AMPs based on hot spot aggregation for the initiation of proteostasis collapse [32]. The main idea is that aggregator peptides have multiple targets in the bacterial proteome and can cause nonspecific aggregation of bacterial proteins and lethal inclusion body formation. The design of peptides based on hot spot aggregation assumed a duplicate APR sequence flanked at the edges of the chain by arginine R-APR-RR to prevent aggregation and increase the absorption capacity of the bacterial cell. In our studies, we constructed a peptide capable of specifically binding to a unique target by the directed co-aggregation mechanism [33,34,35,36]. To do this, we selected APR regions in the target protein sequence using the FoldAmyloid [37], PASTA 2.0 [38], Waltz [39], and AGGRESCAN [40] programs, taking into account the regions predicted by most programs. The design of peptides based on directed co-aggregation involved the synthesis of peptide sequences without modifications, as well as covalently conjugated with cell-penetrating peptides.

The proposed site-based design of APRs is based on years of research on amyloidosis, protein denaturation and renaturation, protein folding, and inclusion body formation [1]. The study of co-aggregation showed that the mutual aggregation of two dissimilar amino acid sequences depends on the identity of the aggregating sequences [41,42,43]. It is reported that the sequence that determines the aggregation of molecules can be quite short, about 5–15 a.a. [2]. Such regions are characterized by enrichment of hydrophobic Val, Leu, Ile, and aromatic Phe, Tyr, Trp amino acids; charged residues Asp, Glu, Lys, Arg, and Pro are often found at the edges of aggregating sequences [44]. Apparently, the formation of “complementary” interactions between polypeptide chains is carried out through the “steric zipper”—two self-complementary β-sheets (regions), giving rise to the spine of an amyloid fibril [45,46,47]. The formation of amyloids is accompanied by a cross-β structure, which is necessary for the “steric zipper” interaction. Khodaparast et al. showed that disruption of β-sheet propensity leads to a decrease in the aggregation and a decrease in the effectiveness of AMPs [32]. In some cases, it has been shown that oligomerization is an important property of AMPs for their functioning [48]. However, it remains not entirely clear whether the formation of the “steric zipper” is the key to the interaction of the peptide and protein through the mechanism of amyloid-like aggregation.

The ability of pathogenic bacteria to form biofilms is one of the key issues that must be addressed to maintain public health. It is known that amyloidogenic proteins promote the primary stage of cell adhesion to the substrate, cell aggregation, and are necessary for the formation of the biofilm matrix [49,50,51,52]. There are also facts confirming the possibility of preventing the formation of biofilms using amyloidogenic antimicrobial peptides. Chen et al. showed that amyloidogenic hexapeptide led to agglutination of bacteria into clusters and thus prevented biofilm formation [53]. It is possible that the mechanism of inhibition of biofilm formation corresponds to the mechanism of bacterial agglutination upon exposure to thanatin.

Thus, understanding the biophysical laws of the formation of amyloid-like structures can help not only to combat amyloid and prion diseases, but can also contribute to the development of a new class of antibiotics, amyloidogenic antimicrobial peptides. 

## 2. Results

### 2.1. Ribosomal S1 Protein Is a Unique Target: Amyloidogenic Properties of S1 Proteins

The choice of a target in a bacterial cell for the co-aggregation process is a very important step in the development of AAMPs. A protein must have a number of important functions for a bacterial cell and, at the same time, not occur in a eukaryotic one. Bioinformatics analysis helped us to find such a protein. It turned out to be the ribosomal S1 protein. The protein is also unique in that the number of domains depends on belonging to one or another phylum, and can only change from 1 to 6 [54]. What is more interesting about this protein is that it represents a repeat of the same S1 domain (having OB-fold structure), which are structurally very similar, and there is variation in the primary structure (Figure 2). It turned out that all Gram-negative bacteria have six domains. The D3 domain for six-domain proteins turned out to be the most conserved. Each domain in this protein has its own function, and these functions are not fully defined. The functional role of domains of S1 from *E. coli* has been studied the most. Thus, it was shown that the D1–D2 domains interact with the 30 S subunit, and the D3–D6 domains interact with RNA [55,56]. D3 domain is of fundamental importance in interaction with mRNA and tmRNA, and is also important in interaction with ribonuclease regB [57]. Moreover, the D6 domain is an autogenous repressor of its own synthesis [58]. 

Ribosomal S1 protein is involved in a number of vital processes, from mRNA recognition and stabilization on the ribosome to gene regulation [60]. For example, the λ phage uses the S1 protein as part of the Qβ replicase [61]. In cells, the S1 protein acts as a regulator of its own synthesis and the biosynthesis of S2 and the EFTs elongation factor [62]. Under stress conditions, the S1 protein is involved in the formation of hibernating ribosomes [63]. There is evidence that the S1 protein is a key target for pyrazinamide in *Mycobacterium tuberculosis* cells [64]. The versatility of this protein makes it a potential target for the development of new antibiotics.

So far, it has not been possible to crystallize a full-length ribosomal S1 protein, but individual domains have been obtained mainly by NMR. It is now possible to predict the structures of these proteins using the AlphaFold 2 program [65]. If we look at the structure of these proteins for different organisms with different numbers of domains, we can see that the protein has an elongated conformation, and the loops between the domains have a helical conformation (Figure 3). We suggest that this structure allows this protein to interact with amyloidogenic peptides and lead to aggregation and the formation of amyloid structures [33,66]. The folded inactive conformation of the ribosomal S1 protein, found in hibernating ribosomes, can probably also bind amyloidogenic peptides, which can prevent the formation of 100 S ribosomes and the death of bacterial cells under stress conditions.

Of course, the question immediately arises, and the protein itself can form amyloid or fibrillar structures. We isolated recombinant forms of these proteins except for *E. coli* and could not select the conditions for their fibril formation, only aggregate forms were obtained [68,69].

### 2.2. Sequence Analysis of Ribosomal S1 Proteins of Model Organisms, Both Strain-Specific and Species-Specific Features of T. thermophilus, E. coli, P. aeruginosa, and S. aureus

We needed to select model and pathogenic organisms for work. For the research, we formulated some requirements for model organisms to test the hypothesis of directed coaggregation. The different degree of pathogenicity of bacteria species potentially helps to understand the role of pre-adaptation of bacteria to the amyloidogenic peptides. The different structure of the cell wall and membrane of Gram-negative and Gram-positive bacteria allows us to evaluate the ability of antimicrobial amyloidogenic peptides to act on the membrane and/or penetrate the membrane. The differences in ribosomal S1 proteins in their ability to aggregate, homology, and in the number of domains potentially allow us to understand the possibility of transferring our strategy to other organisms. In light of these reasons, we decided to select *T. thermophilus*, *E. coli*, **P. aeruginosa*,* and **S. aureus*.*

*T. thermophilus* is a non-pathogenic thermophilic Gram-negative bacterium founded in natural as well as artificial thermal environments and used as a model organism in structural biology [70]. The translation apparatus of the *T. thermophilus* is a subject of on-going research aiming both to discover new antibacterial agents and to understand certain fundamental aspects of translation machinery functioning. Previously, species-specific features of the formation of 100 S hibernating ribosomes in *T. thermophilus* were identified, which is a responsible mechanism under stressful conditions that regulates energy costs [71].

*E. coli* is a Gram-negative bacterium, some strains of which can cause severe community-acquired urinary tract infection, neonatal meningitis, bloodstream infections, and severe intestinal infections [72,73,74]. The pathogenic strains of *E. coli* have a wide range of virulence variants required for colonization and developing infections [75]. A relationship exists between the presence of virulence factors and resistance to ampicillin, cotrimoxazole, and norfloxacin in uropathogenic *E. coli* [76].

*P. aeruginosa* is a widespread Gram-negative bacterium, a human opportunistic pathogen that causes severe respiratory and wound infections especially in immunocompromised patients. *P. aeruginosa* causes 10% of all nosocomial infections, but cases of community-acquired infections caused by this microorganism are increasing. The plasticity and adaptability of the *P. aeruginosa* genome, provided by multiple regulatory genes (about 8% of the genome), allow the pathogen to persist in the host organism for a long time and resist antibiotic treatment [77]. *P. aeruginosa* has a wide range of unique efflux systems. One hundred percent of the strains have at least two drug resistance systems, and 75% of the strains have seven. Resistance to carbapenems was found in 14% of the strains [78].

Staphylococci are the most common natural and normally asymptomatic inhabitants of human skin and mucous membranes. However, as soon as the immune system weakens or the integrity of the skin barrier is violated, staphylococci can cause serious diseases, even death. Various strains of *S. aureus* cause a wide range of hospital infections. Methicillin-resistant *S. aureus* (MRSA) strains are the most common cause of nosocomial infections (HA-MRSA) [79]. Analysis of KEGG modules for 53 *S. aureus* genomes revealed methicillin resistance in 25% of *S. aureus*, 85% of the strains are tetracycline resistant, 45% of the strains have the β-lactam resistance module, and 21% of the genomes carry the QacA efflux system multidrug resistance module [78].

It is known that antibiotic resistance may be associated with the genetic diversity of some bacterial strains [80,81,82]. In silico analysis of the ribosomal S1 protein revealed its high conservatism. The three records found for *T. thermophilus* and six different strains for *P. aeruginosa* from the UniProt database (UniProt release 2022_01) are characterized by a high identity of protein and gene sequences (98–99%). For *E. coli*, 72 different records were found, of which 54 records correspond to proteins containing six structural domains with a length of 557 a.a., and the rest contain a smaller number of structural domains with different protein lengths. Alignment of protein and gene sequences for *E. coli* records containing six structural domains showed their high identity (99%). For *S. aureus* 23 different records with different protein lengths (one record: 103 a.a.; others: 391–400 a.a.) were found. For S1 proteins from *S. aureus* (391–400 a.a.) containing four structural S1 domains, multiple alignment of protein sequences revealed that the percent of identity for some records equals 38%, while most of the records have a high identity in this group (98–100%). Alignment of the gene sequences in this group shows a percentage identity of 53% for some records, for records with high protein sequence identity, the gene identity is 99–100%. Note that in this group there is a strain MRSA252 (UniProt ID: Q6GGT5) in which residue 281Asp is located in the position of the corresponding amyloidogenic region [36] in place of 281Val in the strain MSSA476 (UniProt ID: Q6G987). For strain MRSA252 in the S1 protein sequence, substitution 370Ser is also characteristic in comparison with strain MSSA476, strain N315, strain MW2, and strain Mu50/ATCC700699 (370N). Amino acid in the position 198 (Asp or His) depends on the strain. It should be noted that for each record in the studied dataset, nucleotide sequences of the *rpsA* gene (coding ribosomal S1 protein) were downloaded from the European Molecular Biology Laboratory (EMBL) Nucleotide Sequence Database.

The amino acid composition of S1 proteins from *T. thermophilus*, *E. coli*, **P. aeruginosa*,* and *S. aureus* was analyzed and compared with the corresponding proteome values. As seen from Figure 4, all considered bacterial S1 proteins are enriched with Glu, Asp, and Val and depleted in Ala, Arg, Leu, Met, Phe, Tyr, Cys, and Trp compared to the corresponding proteome values. The distinguishing feature in amino acid composition between S1 proteins from the considered bacteria compared to the proteome values is as follows. In the S1 protein from *S. aureus*, there are more Pro, His and less Asn, Lys. In turn, S1 proteins from *T. thermophilus* are enriched with Thr and depleted in Gly. In the S1 protein from *E. coli* there are less Ser, Ile. The amino acid composition of the S1 protein from *P. aeruginosa* does not differ from the S1 proteins of the other considered bacteria, compared to the proteome values.

### 2.3. Antimicrobial Peptides

Today, more than 3000 AMPs are described in the APD3 antimicrobial peptide database [83]; however, the existing limitations (lack of toxicity/carcinogenicity for the patient, ease of use, and production cost [84]) greatly narrow the range of applicable drugs. Only 8 drugs of peptide nature are approved by the FDA as medicines (Table 1) [85]. These are mainly lipoglycopeptides and cyclic peptides: colistin (it preferentially binds to LPS of Gram-negative bacteria and disrupts membranes [86]), vancomycin, daptomycin, oritavancin, telavancin, teicoplanin, and dalbavancin. Among the AMPs used in medical practice, only gramicidin is a linear peptide. The main mechanism is the ability of dimers to form channels in the cell membrane, but its use is limited to topical application due to toxicity when administered intravenously.

The smallest known AMP consisting of 7 amino acids (Lys-Val-Phe-Leu-Gly-Leu-Lys) was isolated from *Jatropha curcas* [93]. This cationic antimicrobial peptide is active against a variety of pathogenic bacteria: *Salmonella typhimurium* ATCC 50,013, *Shigella dysenteriae* ATCC 51,302, *Pseudomonas aeruginosa* ATCC 27,553, *Staphylococcus aureus* ATCC 25,923, *Bacillus subtilis* ATCC 23,631, and *Streptococcus pneumoniae* ATCC 49,619. This peptide killed microbes mainly by destroying their cell walls and membranes.

Among antimicrobial peptides, defensins are the best characterized. Defensins are cationic peptides up to 60 a.a., cysteine-rich, forming α/β motifs with a broad spectrum of antimicrobial activity [94,95]. It is known that defensins have a membrane-active effect. Some membrane lipids promote oligomerization of defensin–lipid complexes [96,97]. Oligomerization is necessary for the formation of a membranolytic complex. Interestingly, defensins may specifically interact with certain membrane lipids and thus may have specificity for certain pathogens [94]. Human α-defensin 6 (HD6) and β-defensin 1 (HBD-1) have a non-membrane antimicrobial effect; nanonetworks are formed by oligomerization, which prevents the spread and reproduction of bacteria [3,98,99]. The need for oligomerization for anti-microbial activity highlights the importance of the structure that peptides are able to form.

A possible ring-like oligomeric structure of plant antifungal *Nicotiana alata* defensin 1 (NaD1) bound to phosphatidylinositol 4,5-bisphosphate (PIP2) is shown in Figure 5. This structure consists of 24 NaD1-PIP2 units. The inner and outer diameters of this structure are about 30 and 100 Å, respectively. For the construction of this ring-like oligomer, the structure 4CQK from the Protein Data Bank was taken as a template.

Peptides from some plant antifungal defensins can form amyloid fibrils. It was shown that the NaD1-19 peptide does not form amyloid fibrils compared to the RsAFP-19 peptide (19 amino acid fragment from the C-terminal region of *Raphanus sativus* antifungal protein). Thus, the ability to form amyloids is not a general property of peptides from plant defensins that have antifungal features [101].

### 2.4. Physicochemical Properties of AMPs 

The advantage of AMPs over antibiotics is their multiple spectra of action. For example, permeabilization (membrane permeability) of the bacterial cell membrane, inhibition of receptor membrane proteins (channels), and co-aggregation with the bacterial proteome. Another advantage of AMPs is its antimicrobial effect combined with biodegradability and generally low toxicity, which is important for use in clinical practice, the food industry, and agriculture.

To construct a matrix and evaluate the correlation distribution between the physicochemical properties of antimicrobial peptides and amyloidogenicity (AM), we evaluated a sample from the AMP dataset in 39,833 sequences [102]. The number of selected antimicrobial peptides is justified by varying the amino acid sequences (length) in order to determine the frequencies of occurrence of hot spots, i.e., amino acids contributing to the amyloidogenicity of the antimicrobial peptide. Then, from the analyzed dataset, AMPs were stochastically clustered (Figure 6A) using the neural network prediction method (APPNN, amyloid propensity prediction neural network) and divided into two sub-clusters: AMPs that are prone to amyloidogenicity (19,913 sequences with a correlation cut-off coefficient greater than 0.6) and AMPs that are not prone to amyloidogenicity (19,920 sequences with a correlation cutoff less than 0.6).

It should be noted that neural network clustering was performed taking into account the correlating physicochemical properties of AMPs, such as aliphatic index (α), instability index (II), hydrophobicity (H), charge (Z), Boman index (IB), antimicrobial peptide length (L), positively charged a.a. (a.a.(+)), negatively charged a.a. (a.a.(−)). This was undertaken in order to analyze the contribution of these physicochemical properties to amyloidogenicity.

According to the results of the correlation distribution between physicochemical parameters, values such as hydrophobicity (H) and the aliphatic index of amino acid side groups (α) correlate well with a correlation value of 0.67 (Figure 6A). An increase in hydrophobicity leads to a decrease in specificity with respect to the bilipid layer of the pathogen and to “poor” solubility of the peptide in the polar phase. As for the Boman index (IB), which is responsible for the ability of the peptide to bind to the bacterial membrane [103], it correlates with the instability index (II) [104] (the correlation coefficient was 0.59). The Boman index function calculates the potential interaction index (affinity) and solubility of hydrophobic and hydrophilic amino acid residues in the bilipid layer.

It was found that with an increase in the length of the peptide, the number of both positively charged a.a. (+) and negatively charged a.a. (−) residues increases, which correlates with the length of AMP (L). Perhaps, with an increase in positively charged a.a. (+) the affinity (selectivity) of the action of AMPs decreases and, accordingly, the reversible and irreversible antimicrobial effect may decrease due to amphipathicity or screening of the positive charge in the molecular structure of the peptide. It is worth paying attention to the lack of correlation between the peptide length (L) and charge (Z) in the correlation matrix.

Figure 6B,C show the correlation matrices of the distribution between the physicochemical properties of AMPs and AMPs, both prone and not prone to amyloidogenicity, respectively. According to the results of the correlation distribution between the physicochemical properties of AMPs and AMPs, which are prone to amyloidogenicity (Figure 6B), physicochemical properties, such as amyloidogenicity (AM) and peptide length (L), slightly correlated, amyloidogenicity (AM) and the amount of positive a.a.(+) and negatively charged a.a.(−) amino acids. A slight correlation in this case can be interpreted by the heterogeneity of the dataset and the distribution of positive and negative charge in the AMPs. However, when comparing the correlation distributions between the physicochemical properties of AMPs and AMPs that are not prone to amyloidogenicity (Figure 6C), apart from the correlation of both amyloidogenicity (AM) and peptide length (L), amyloidogenicity (AM) and the number of negatively charged amino acids a.a.(−), it can be seen that amyloidogenicity (AM) and aliphatic index (α), instability index (II), hydrophobicity (H), charge (Z), and Boman index (IB) are weakly correlated. 

High anticorrelation is observed between instability index (II) and Boman index (IB) for all three cases.

### 2.5. Amyloidogenic Peptides

It is known the amyloidogenic peptides (Aβ(1-42), serum amyloid A, microcin E492, temporins, PG-1) exhibit antimicrobial activity against bacteria [105,106,107], and fungi, and viruses [108,109]. Often antimicrobial amyloidogenic peptides are active at a concentration of about ~10–20 μM (30 μM temporin [110], 4–50 μM peptides on based PG-1 [111]). These facts suggest a connection between their antimicrobial properties and cause aggregation and the formation of amyloid-like fibrils [9,112].

When developing sequences for amyloidogenic peptides, we also considered other properties of the ribosomal S1 protein to be important. Homologous aggregation is preferential for amyloidogenic peptides and proteins, so the cellular target should only be present in the target organisms. The ribosomal S1 protein is present only in bacterial proteomes. The amyloidogenic antimicrobial peptides developed by us should specifically interact with the ribosomal S1 protein by the mechanism of directed co-aggregation [67].

Based on the idea that peptide self-aggregation may be a common theme for the mode of action of antibacterial peptides, we used four popular predictive tools to detect amyloidogenic regions in ribosomal S1 protein from four organisms (Figure 7). 

After analyzing the consensus regions and the amino acid composition of these regions, we selected for synthesis peptides 10 amino acid residues long, as for the previously studied amyloidogenic peptides from the Aβ peptide and Bgl2 protein from the yeast cell wall [113,114]. In total, 19 amyloidogenic peptides were synthesized and for all of them their ability to form fibrillar structures was tested using the fluorescence intensity of thioflavin T and electron microscopy [59,69]. It turned out that almost all peptides synthesized by us form fibrils or pre-fibrils.

### 2.6. Antibacterial Activity of Hybrid Amyloidogenic Peptides

The peptide design we follow includes three functional parts: a cell-penetrating peptide (CPP), a linker, and an amyloidogenic region. The role of the amyloidogenic region was described above. CPPs are usually amphipathic or cationic oligopeptides capable of transporting attached cargo across cell membranes. The positive charge allows CPP to impart a net positive charge to the entire peptide molecule, which is important for interacting with the bacterial membrane and stimulating the uptake of the peptide into the cell. We used a fragment of the HIV-1 transcription factor TAT having the amino acid sequence RKKRRQRRR. One of the problems of CPP is their biological activity. A number of authors note that Tat-peptide exhibits antimicrobial activity against *E. coli*, *S. typhimurium*, *B. subtilis*, *S. epidermidis*, and *S. aureus* [115]. However, in our studies, Tat-peptide showed no antimicrobial activity against *E. coli*, *S. aureus*, and *P. aeruginosa* [35]. The linker allows creation or prevention of the formation of the secondary structure of the peptide. In our studies, we used the GGSarG or GGGG sequence. The sequence of four glycines can contribute to the flexibility of the polypeptide chain, while the presence of sarcosine makes the structure rigid. In addition, modified peptides can provide a longer duration of action, as they are more resistant to proteases. Our study showed that Sar-modified peptides exhibit a wide range of antimicrobial activity against MRSA, *P. aeruginosa*, *E. coli*, and *B. cereus* [36]. As a result of the series of experiments, the minimum inhibitory concentrations (MICs) of the modified peptides were determined against strains of pathogenic methicillin-resistant *S. aureus* (MRSA), *S. aureus*, and *P. aeruginosa* (Table 2) [35,36]. 

As follows from Table 2, we found that the antimicrobial effect of the same peptides against various pathogenic microorganisms is comparable to the MIC (~1–5 µM) of the commercial antibiotic gentamicin sulfate [116,117].

It is important that some peptides demonstrated high antimicrobial activity against pathogenic strains of MRSA, *S. aureus*, and *P. aeruginosa*. Peptides R23F, R23DI, and R23EI were designed based on the amyloidogenic sequences of ribosomal S1 protein from *S. aureus*, so they are more effective against strains of *S. aureus* or MRSA, and their lower activity against *P. aeruginosa* was expected. The peptides R23F, R23DI, and R23EI proved to be the most effective against the ATCC 43,300 MRSA strain in liquid medium: MIC 3 μM for the R23F peptide, MIC 3 μM for R23DI, and 6 μM for the R23EI peptide. Low MIC values (at the level of commercial antibiotics gentamicin sulfate) were also determined for R23F, R23DI, and R23EI against *S. aureus* strain 209P in liquid medium—MIC 3 µM for R23F peptide, MIC 6 µM for R23DI, and MIC 12 µM for peptide R23EI. The MICs of the peptides R23F, R23DI, and R23EI against the *P. aeruginosa* ATCC 28,753 strain in liquid medium were 12, ≥12, and >12 µM, respectively. In general, the peptides demonstrated high antimicrobial activity against *P. aeruginosa* and *S. aureus* strains grown in liquid medium and lower antibacterial activity in tests on agar medium. The R23R and R23L peptides were able to inhibit the growth of *P. aeruginosa* strain ATCC 28,753 cells at concentrations of 6–12 µM (MIC = 12 µM), but not the growth of *P. aeruginosa* strain PA103 cells (MIC > 12 µM).

### 2.7. Potential Limitations and Benefits of Using AAMPs

It is known that amyloidogenic peptides can stimulate the formation of amyloid fibrils both in in vitro and in animal models [42,118,119]. Genome analysis shows that a significant proportion of proteins have evolutionarily conserved regions capable of aggregation [120,121]. The ability of proteins to aggregate may be necessary for the performance of physiological functions. Aggregate-capable regions form the hydrophobic core of the protein and contribute to the correct folding of the molecule [122]. Additionally, such sites are involved in protein–protein interactions, which are necessary for the formation of non-membrane compartments involved in both metabolic reactions and stress reactions [123]. This means that it is necessary to take into account the level of amyloidogenicity and selectivity of action of amyloidogenic peptides in order to reduce the potential toxic effect of the antimicrobial amyloidogenic peptide due to (i) “non-functional” aggregation and (ii) “functional” aggregation [120]. It is known that the problem of toxicity of antimicrobial peptides is solved by modifying amino acids; however, this reduces the membrane-active effect against eukaryotic cells [124]. What should be taken into account when designing a peptide to prevent binding to eukaryotic intracellular proteins? It is probably necessary (iii) to identify aggregation sites common to eukaryotic and prokaryotic proteins; (iv) to determine the physicochemical characteristics that regulate the degree of aggregation between proteins; (v) to determine the differences in the mechanisms of maintenance of proteostasis in eukaryotic and prokaryotic cells and the range of tolerance when exposed to amyloidogenic peptides; and (vii) to exclude APRs that have the potential to stimulate amyloid production *in vivo*. 

Despite the potential limitations in the use of amyloidogenic peptides, it is important to consider their potential benefits. A critically important advantage of AAMPs is the ability of peptides with varying degrees of specificity to bind to the target. This is a unique feature, as it is quite easy to synthesize peptides with targeted action on one or more targets. Additionally, the possibility of changing the peptide template makes it possible to introduce various modifications that modulate antimicrobial activity, tissue specificity, toxicity, and other types of biological activity. Note that the peptides synthesized by us are non-toxic for human fibroblast cells [35,36]. The greater resistance of amyloidogenic sequences to proteolysis is an additional advantage of amyloidogenic antimicrobial peptides, due to the fact that they are more stable.

## 3. Conclusions

The strategy of creating peptides that induce bacterial protein aggregation appears to merit further attention and further research. However, potential problems should be considered as well. When searching for regions of the proteome of microbial organisms prone to aggregation, it is also necessary to analyze the presence of similar amino acid sequences in the human body. It is therefore possible to assume in advance the possibility of fibril-forming effects of AAMP on the human body.

In our previous work, we investigated the ribosomal S1 proteins. With the help of bioinformatics approaches, it was found that the S1 protein contains in its sequence regions that tend to aggregate and form fibrils, and in some domains their percentage is higher than in the whole protein. Potentially, these regions can become targets for directed aggregation. It has been suggested that peptides built on the basis of these regions can aggregate with the S1 protein and cause its further aggregation. It was found in vitro experiments that the synthesized peptides are capable of stimulating fibril formation of the S1 protein. To increase the ability of the peptide to penetrate through the bacterial membrane, the TAT fragment was used, and to increase the stability of the peptide, nonstandard amino acids (sarcosine) were inserted into the linker part between the TAT fragment and the amyloidogenic site. Further experiments on the effect of peptides with this design on bacterial cultures demonstrated the presence of antibacterial properties in the peptides we developed.

In the previous work, our team noted that the presence of strong amyloidogenic properties in the peptide does not mean that the peptide has intense antimicrobial properties [35]. It has not yet been clarified which peptide structures are the main factor influencing the bacterial cell. If these are the peptide monomers, it is likely that the ability to rapidly aggregate and form oligomers may prevent the peptide from entering the bacterial cell and reduce its antimicrobial activity. However, while the full mechanism of the effect of AAMP on bacterial cells has not been elucidated, it is also worth considering possible additional mechanisms of action on the bacterial cell, such as the incorporation of peptide oligomers into the cell wall with further formation of pores. Despite the presence of the proposed mechanism of action, it is not yet possible to accurately assert its correctness. In the future, in order to identify and describe the mechanism of action of AAMP, additional experiments should be carried out with the study of bacterial proteomes exposed to peptides. Additionally, the use of fluorescently labeled peptides can reveal the main sites of action of peptides inside cells.

## Figures and Tables

**Figure 1 ijms-23-05463-f001:**
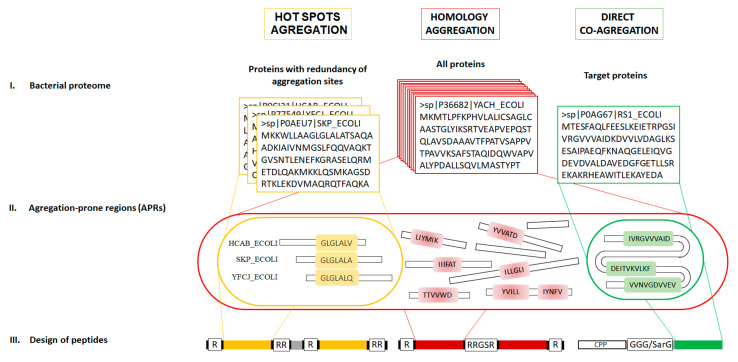
Strategies for the development and design of amyloidogenic antimicrobial peptides include three steps: selection of a list of protein sequences, search in the list of APRs, and construction of a peptide. The homologous aggregation (red color) strategy [1] involves searching for APRs in the entire set of bacterial proteins. A peptide designed in this way consists of a duplicate section of APRs separated by a glycine–serine linker and flanked by arginine and lysine to maintain colloidal stability. The hot spot aggregation (yellow color) strategy [32] involves searching for the most common APRs in the bacterial proteome. A peptide designed in this way represents a duplicated section of APRs flanked by arginine at the ends of the sequence and separated by a proline linker. The direct co-aggregation (green color) strategy involves the preliminary selection of a target protein and the search for APRs in it. An amyloidogenic site conjugated to a CPP sequence separated by a GGSarG or GGGG type linker.

**Figure 2 ijms-23-05463-f002:**
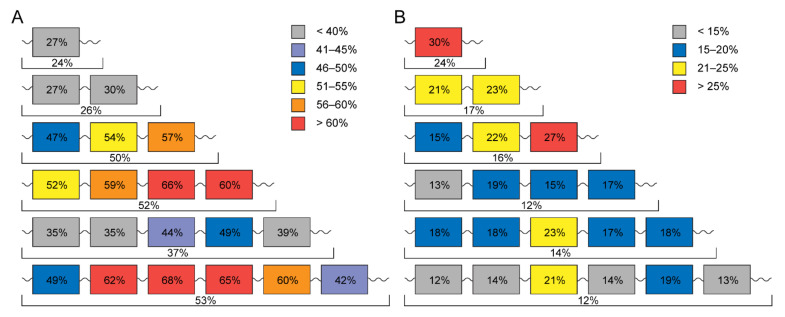
Average percentage of identity (**A**) and amyloidogenic regions predicted by the FoldAmyloid program [37] (**B**) within each domain as well as all domains in proteins containing different numbers of domains are given. A dataset containing 1331 records of ribosomal S1 proteins was used [59].

**Figure 3 ijms-23-05463-f003:**
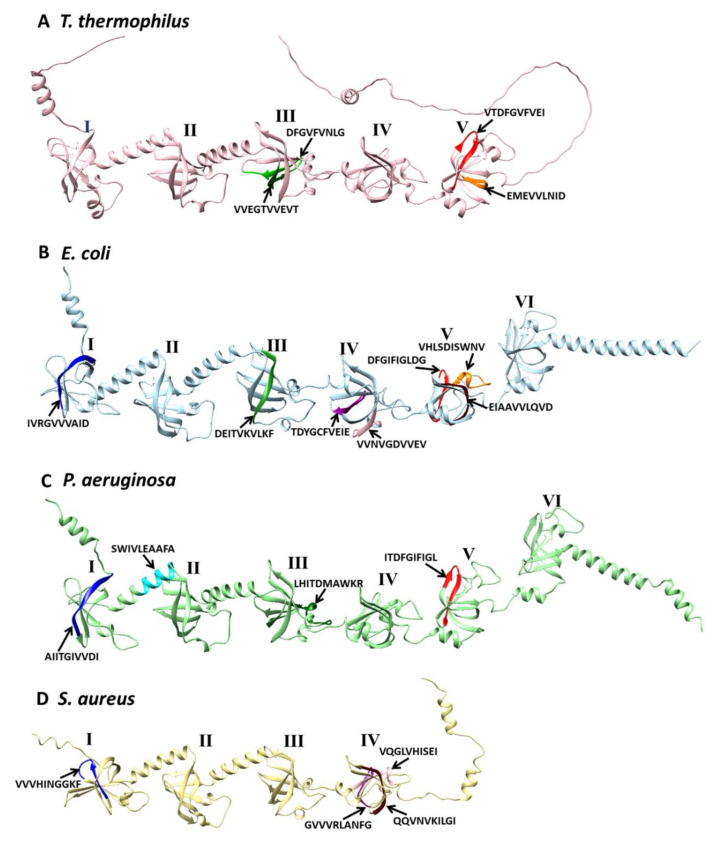
Structures of S1 proteins from *T. thermophilus* (**A**), *E. coli* (**B**), *P. aeruginosa* (**C**), and *S. aureus* (**D**) predicted by AlphaFold 2 [65]. The predicted and synthesized consensus amyloidogenic regions by four programs are indicated on each structure. Visualization of structures was carried out using the UCSF Chimera program [67].

**Figure 4 ijms-23-05463-f004:**
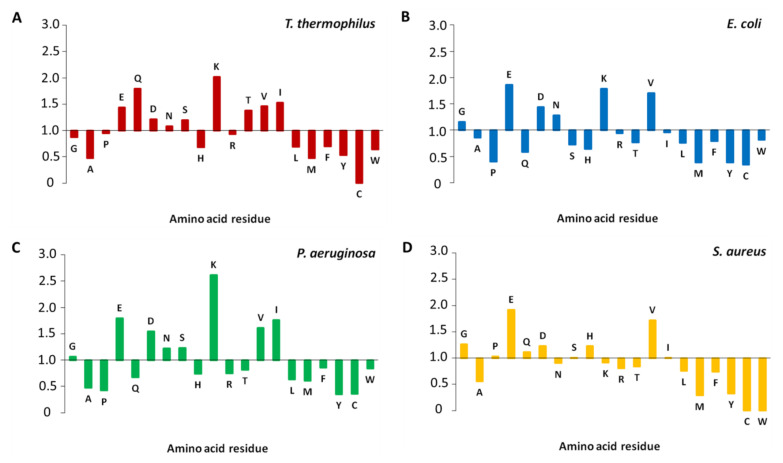
Comparison of the amino acid composition of the structures of S1 proteins from *T. thermophilus* (**A**), *E. coli* (**B**), *P. aeruginosa* (**C**), and *S. aureus* (**D**) with the corresponding bacterial proteome values.

**Figure 5 ijms-23-05463-f005:**
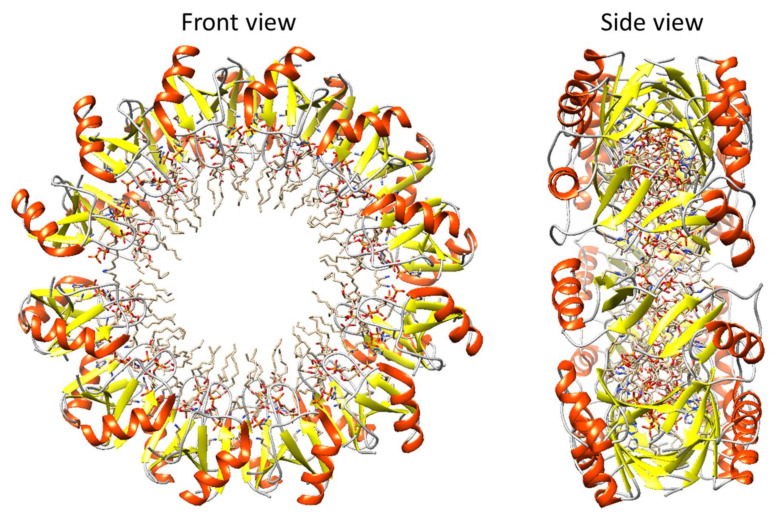
Constructed oligomeric structure of plant antimicrobial peptide defensin NaD1 containing 24 NaD1-PIP2 units. Packing of oligomers was completed by the YASARA program [100].

**Figure 6 ijms-23-05463-f006:**
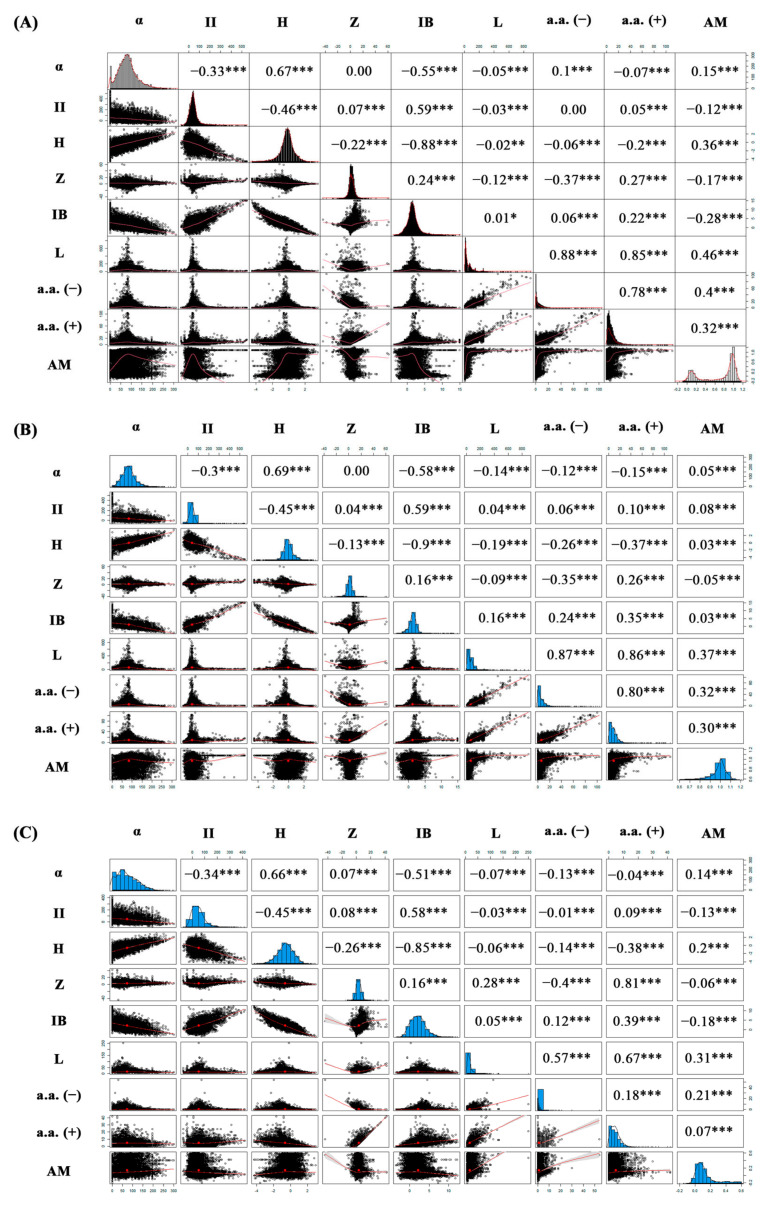
Correlations between physicochemical properties of AMPs and AMPs prone to amyloidogenicity (AM) (**A**). α is aliphatic index, II is instability index, H is hydrophobicity, Z is charge, IB is Boman index, L is antimicrobial peptide length, a.a. (+) is positively charged a.a., a.a. (−) is negatively charged a.a., AM is amyloidogenicity. The result of a stochastic cluster analysis of AMPs prone to amyloidogenicity (AM). The correlation matrices of the distribution between the physicochemical properties of AMPs and AMPs, both prone and not prone to amyloidogenicity, respectively, (**B**) and (**C**). The calculations were performed in the R package. Symbol “*” means the most significant correlations between properties. The greater the number of “*”, the more significant the correlation between the physicochemical properties of AMPs and amyloidogenicity (AM).

**Figure 7 ijms-23-05463-f007:**
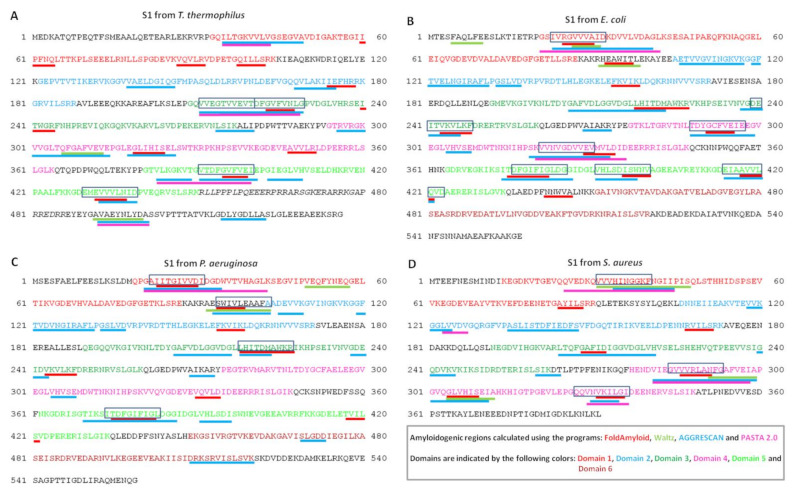
Predicted amyloidogenic regions for ribosomal S1 protein from four organisms *T. thermophilus* (**A**), *E. coli* (**B**), *P. aeruginosa* (**C**), *S. aureus* (**D**) using four programs: FoldAmyloid, Waltz, AGGRESCAN, and PASTA 2.0. The synthesized peptides are circled in a rectangle.

**Table 1 ijms-23-05463-t001:** List of AMPs approved by the FDA as medicines.

Name of AMP	Structure of AMP	Mechanism of Action	Reference
Gramicidin	Linear, forms a spiral	Pore formation from dimers	[87]
Colistin	Cyclic lipopeptide	Membrane-lytic peptide	[88]
Daptomycin	Cyclic lipopeptide	Membrane-lytic peptide	[89]
Vancomycin	Lipoglycopeptide	Inhibitor of cell wall synthesis	[90]
Oritavancin	Lipoglycopeptide	Dual-mechanism: membrane-lytic peptide and inhibitor of cell wall synthesis	[91]
Dalbavancin	Lipoglycopeptide	Inhibitor of cell wall synthesis	
Telavancin	Lipoglycopeptide	Dual-mechanism: membrane-lytic peptide and inhibitor of cell wall synthesis	
Teicoplanin	Lipoglycopeptide	Inhibitor of cell wall synthesis	[92]

**Table 2 ijms-23-05463-t002:** MIC for hybrid peptides.

Sequence of Hybrid Peptides and Folding Patterns Predicted by AlphaFold 2	Strain of the Pathogenic Microorganism	MIC for the Tested Hybrid Peptide (µM)
Based on the sequence S1 protein from *P. aeruginosa*
RKKRRQRRRGGGGITDFGIFIGL 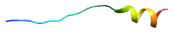	MRSA strain ATCC 43,300 (resistant to ampicillin) *S.*	12
*aureus* strain 209 (resistant to aztreonam)	>12
*P. aeruginosa* (strain ATCC 28,753) (resistant to sulfamethoxazole)	12
RKKRRQRRRGGSarGLHITD-Nle-AWKR 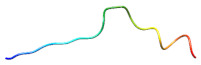	*P. aeruginosa* (strain ATCC 28,753) (resistant to sulfamethoxazole)	12
*P. aeruginosa* (strain PA 103) (resistant to levomycetin)	>12
RKKRRQRRRGGSarGITDFGIFIGL 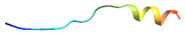	*P. aeruginosa* (strain ATCC 28,753) (resistant to sulfamethoxazole)	12
*P. aeruginosa* (strain PA 103) (resistant to levomycetin)	>12
Based on the sequence S1 protein from *S. aureus*
RKKRRQRRRGGSarGVVVHI-Asi-GGKF 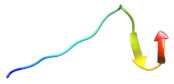	MRSA strain ATCC 43,300 (resistant only to ampicillin)	3
*S. aureus* strain 209 (resistant to aztreonam)	3
*P. aeruginosa* strain ATCC 28,753 (resistant to sulfamethoxazole)	12
RKKRRQRRRGGSarGLTQFGAFIDI 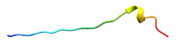	MRSA strain ATCC 43,300 (resistant only to ampicillin)	3
*S. aureus* strain 209 (resistant to aztreonam)	6
*P. aeruginosa* strain ATCC 28,753 (resistant to sulfamethoxazole)	12
RKKRRQRRRGGSarGVQGLVHISEI 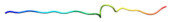	MRSA strain ATCC 43,300 (resistant only to ampicillin)	6
*S. aureus* strain 209 (resistant to aztreonam)	12
*P. aeruginosa* strain ATCC 28,753 (resistant to sulfamethoxazole)	>12

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
