# Peer review of "Amyloidogenic Peptides: New Class of Antimicrobial Peptides with the Novel Mechanism of Activity"

_ijms, 2022, doi:10.3390/ijms23105463_

Round 1

Reviewer 1 Report

Protein/peptide aggregation is an ubiquitous property that is implicated in many biological processes including the formation of amyloids and the activities of antimicrobial peptides (AMPs). Studies have shown the common properties of the mechanism of AMPs and amyloid aggregation, and their ability to disrupt the lipid membrane. As a result, certain peptides have been shown to exhibit both antimicrobial and amyloid activities. In this review, the authors cover the studies on the antimicrobial effects of the designed peptides on two model organisms, E. coli and T. thermophilus, and two pathogenic organisms, P. aeruginosa and S. aureus.  This is an important and growing area, and therefore the review article is timely. Overall, this review article is well written and can be published after a careful revision. 

Suggestions for revision are given below: 

1) Although the authors restrict the focus of the review article to the antimicrobial effects of the designed peptides on two model organisms and two pathogenic organisms, it is important to expand the Introduction to briefly cover previous studies on AMPs and amyloid peptides.

2) One of the very important AMPs that forms fibrils is LL-37, the only human cathelicidin derived AMP. A brief mention would be useful. 

3) A brief mention of the common modes of membrane disruption by AMPs and amyloid peptides would be useful. Include the following refs: 

  • Fatafta et al Biophysical Chemistry26 October 2021 
  • Biophys J. 2012 Aug 22;103(4):702
  • Halder et al "Biophys.Chem.282, 106759 (2022)"

3) The following important articles related to the topic are useful to include in this Review. 

  • Nguyen et al Chem.Rev. (2021)
  • Kotler et al ChemSocRev 2014, 43:6692
  • Cawood et al "Vizualizing and trapping transient ..." 

Reviewer 2 Report

In this manuscript, the author overviews the exploitation of amyloidogenic antimicrobial peptide (AAMP) they developed using S1 ribosomal proteins, describes the development of these peptides in protein bioinformatics analysis, and advocates the role of amyloidogenic antimicrobial peptides in the field of antimicrobial peptides.

The amyloidogenic antimicrobial peptides bind to pathogen surface receptors through a co-aggregation mechanism to produce a “protein silencing” effect. A series of amyloidogenic antimicrobial peptides developed by the authors have good bactericidal activity against drug-resistant bacterial strains.

Major comment:

  1. This manuscript is the first review article on amyloid antimicrobial peptides (AAMPs) and contributes to the discussion of the development of such peptides. However, the content of this manuscript is roughly in the same direction as the author’s paper published in IJMS this year (2022, Kravchenko), emphasizing the amyloidogenic antimicrobial peptide series and development method developed by the authors. The manuscript seems to expand the introduction and conclusion of the previous published literature, and ends with the applicability and challenges to be overcome for amyloidogenic antimicrobial peptides.
  2. In the concept of peptide design, the authors should supplement the relationship between antibacterial activities and amyloidogenic activities of antimicrobial peptides and which amino acid or sequence is important for both activities.
  3. In Figure 6, the authors described the physicochemical parameters of antimicrobial peptides instead of amyloidogenic peptides. The correlations between physicochemical parameters for amyloidogenic peptides should be conducted and discussed in this manuscript.

Minor comment:

  1. On page 2, please check the form for all the noun “co-aggregation” for consistency.

Round 2

Reviewer 2 Report

The authors have answered all of my questions.